# Explainability Via Causal Self-Talk

**Nicholas A. Roy**[*]
DeepMind
nroy@deepmind.com

**Junkyung Kim**[*]
DeepMind
junkyung@deepmind.com

**Neil Rabinowitz**[*][†]
DeepMind
ncr@deepmind.com

## Abstract

Explaining the behavior of AI systems is an important problem that, in practice, is generally avoided. While the XAI community has been developing an abundance of techniques, most incur a set of costs that the wider deep learning community has been unwilling to pay in most situations. We take a pragmatic view of the issue, and define a set of desiderata that capture both the ambitions of XAI and the practical constraints of deep learning. We describe an effective way to satisfy all the desiderata: train the AI system to build a causal model of itself. We develop an instance of this solution for Deep RL agents: Causal Self-Talk. CST operates by training the agent to communicate with itself across time. We implement this method in a simulated 3D environment, and show how it enables agents to generate faithful and semantically-meaningful explanations of their own behavior. Beyond explanations, we also demonstrate that these learned models provide new ways of building semantic control interfaces to AI systems.

## 1 Introduction

As modern machine learning systems become more powerful and embedded in our lives, the need to have these systems explain their behavior becomes increasingly urgent. Despite incredible performance in a variety of domains, almost all systems are completely unable to provide a satisfying answer to the simple question, "Why did you do that?".

While there have been innovations in particular explainability methods [e.g. 8, 55, 56, 66], and some attempts to apply techniques to domains that demand it [e.g. 23, 65], the vast majority of deep learning applications do not engage with these techniques. This is in part because there are few general methods, and most incur costs such as reduced performance and scalability; compounding this, existing techniques often fail to deliver a useful or credible account of the systems' behavior.

We believe that the reason for this is that explanation systems tend to fail to satisfy at least one of five key desiderata discussed in detail below, namely: groundedness, flexibility, minimally-interfering, scalability, and faithfulness. Two particular failure modes are most common: the addition of an explanation system limits generality or hurts performance (e.g. when it requires explicitly structuring the base system); or the explanations it produces cannot be trusted to be accurate (e.g. with many external or post-hoc methods). An ideal explanation system would avoid both these pitfalls.

In this paper, we propose one such solution. We take a pragmatic viewpoint that providing explanations of a system's behavior is, in essence, a task of building a *model* of that system. We argue that the AI system itself is well-positioned to supply a model that fulfills all the aforementioned desiderata. We thus train the base system to supply a "self-model" alongside its main representations. Importantly, the base system and self-model are subject to different constraints: we want the base system to be primarily optimized for performance, while we want the self-model to take on a faithful,

---

[*]Equal contribution
[†]Corresponding author

36th Conference on Neural Information Processing Systems (NeurIPS 2022).

semantically-interpretable form. The different form allows the self-model to deliver supplementary utility that the base system does not, e.g. providing an interface to external users, and enabling them to understand, predict, and control the base system in convenient ways.

Our contributions are as follows. In Section 2, we articulate five key desiderata for explanation systems. However, there is some tension between these. In Section 3, we describe how common historical XAI techniques resolve these trade-offs by making extreme choices between the desiderata. Section 4 marks a transition from discussing the desiderata in theory to exploring them in practice, as we demonstrate that a balance can be found between the five desiderata by training AI systems to build self-models. In particular, we introduce a solution concept for Deep RL agents: Casual Self-Talk (CST), in which an agent must learn to use its own outputs as inputs. In Sections 5-6 we show that CST allows model-free RL agents in an embodied 3D virtual environment to faithfully explain their behavior in terms of semantically-accessible beliefs about the state of the world.

## 2   Desiderata for explanatory models

Our goal is to build an explanatory model of an AI system, i.e. a mechanism that translates between its internal representations and/or computations, and some other representational form. We set out five key desiderata for such a model. We flesh these out in more detail in Section 3.

**Grounded.**   For the model to be useful to external users, it should deploy a representation language that is semantically-grounded. This may require additional data to train.

**Flexible.**   The method should be able to produce models of many different representational forms (e.g. classes, graphs, strings). Ideally it should also be sufficiently general to be agnostic to the base system's input/output modalities, and possibly even its overall architecture.

**Minimally-interfering.**   Training and inference on the model should have as little impact as possible on the training and inference of the base system. Solutions which are disruptive to performance are highly unlikely to be adopted in practice.

**Scalable.**   Once a solution is found, it is straightforward to apply it to any size network.

**Faithful.**   The model should provide accurate descriptions of the underlying system. The gold standard is a *causal model* of the agent, which facilitates validation of its accuracy through interventions [38]. This is possibly the most challenging to achieve, and is discussed further in the specific context of decoders in Section 3.

A number of past works have also articulated desirable properties for explanations [e.g. 28, 35, 36, 38, 49, 53, 69].[3] Our desiderata focus less on the qualities which constitute a good explanation per se, but instead on what is desired of the method itself that generates the explanation.

## 3   Common historical solutions

We examine whether various existing explanation techniques satisfy each of these desiderata.

**Do nothing.**   In standard Deep Learning (DL), we take the base system's learned internal representations at face value. Using the base system as a model of itself clearly has benefits of being minimally-interfering (no changes are made), scalable (as much as DL itself), and faithful (since the internal representations are causal to the base system's output). However, the approach entirely sacrifices grounding and flexibility.

**Attribution methods.**   A set of popular techniques attempts to model a neural network's decision-making at the input level, by estimating contributions to its output from input features [e.g. 18, 27, 56], or training examples [e.g. 6, 14, 42, 43, 46, 71], or by finding optimal inputs for a given output [e.g. 55, 73]. Such approaches benefit from being minimally-interfering (they are entirely post-hoc) and scalable (they can be automated). However, some methods have been empirically shown to be

---

[3]Note that some terms, such as 'faithfulness', do not have standard definitions across this literature [35].

unfaithful [2, 7], while methods which explicitly aim for faithfulness such as LIME [13] and Anchors [60] are only local in scope, and are expensive to scale. Most importantly, all such methods sacrifice flexibility: one is constrained to produce explanations of a very particular form, e.g. expressed in terms of the input modality, whose interpretation may be highly subjective.

**Explicitly structuring networks.** A sub-field of deep learning has emerged wherein the intermediate computations are constrained to particular forms, and/or are grounded by external data [e.g. 5, 10, 16, 23, 25, 47, 62, 63, 72]. As explanations of behavior, structured intermediates are attractive as they are both grounded and faithful by design. However, this is achieved at the expense of minimal-interference, as it requires fundamentally altering the base system to explicitly depend on this structure. It is also limited in flexibility and scalability, as the technique can also only be applied when sufficient domain knowledge or data is available, and only in the form dictated by these prior constraints. These issues are exacerbated by the fact that the representations required to optimize performance objectives are often at odds with those that would subserve explainability.

**Fine-grained mechanistic interpretability.** One common approach is to allow a base system to train and run without interference, and to post-hoc dissect its representations or computations, often using techniques borrowed or inspired from neuroscience [e.g. 17, 21, 26, 57]. Such approaches champion the minimal-interference desideratum. In the best case, they promise to uncover faithful descriptions of the actual causal processes operating within the base system. However, this comes at a huge cost to scalability. Many human hours are typically expended in reverse engineering, often yielding results that are limited to a handful of neurons or circuits seen in a single network. Moreover, by making strong commitments to respect the internal representations of the base system, it is very difficult to ground the models that arise, let alone to flexibly choose the form they take.

**Decoders.** Decoders can be viewed as trained tools to map from the internal private language of a base system to an interpretable representation space [e.g. 4, 11, 15, 31–33, 44, 52, 58, 74]. This approach has many attractive qualities: it produces semantically-grounded representations of the base system's internal private language; is flexible in the choice of modality; scales well; and is minimally-interfering, insofar as decoders can be separated from the inference path of the main computational graph and gradients can be stopped from propagating to the base system if desired (though are often not to assist representation learning, [e.g. 3, 37, 40, 48, 54]). This is all achieved, however, at the expense of faithfulness, as decoded outputs are off the causal path of the base system [47]. Decoders just report whether information is *present* in the source representation, not whether that information is actually being *used* for behavior [12, 20, 59]. Similar faithfulness issues arise with post-hoc behavioral analyses [e.g. 19, 30, 39, 50, 66, 70].

## 4 Solution concept

The challenges faced by previous attempts at explainability show that the tensions between the desiderata are considerable. It is nonetheless possible to reason a way forward.

**Motivation.** First, we want a description of the base system that is both grounded and flexible. The internals of the base system are neither. While we could explicitly structure the base system to change this, this would violate the minimal interference desideratum. Thus we need *a model* of the base system. Second, the solution must be scalable. This means we cannot rely on human labor to map from the base system's representations to the model's representations. The only scalable approach is to *learn the model*. Third, the solution must be faithful. Our gold standard of faithfulness requires the ability to validate the model by intervening on it. To do this, one must be able to map in the opposite direction: from interventions on the model onto interventions on the base system. Because both the base system and model are learned, this mapping between them must also be learned. The most direct way of learning this mapping from model to base system is to *feed the model's outputs as inputs to the base system*.

A solution to these problems is thus a "self-model" which is intertwined with the base system. We develop an instance of this solution in a particular setting: a Deep RL agent, acting in an episodic POMDP, which builds a model of itself in terms of a set of semantically-grounded beliefs about the state of the world. We consider how to expand the scope of this technique in Section 7.

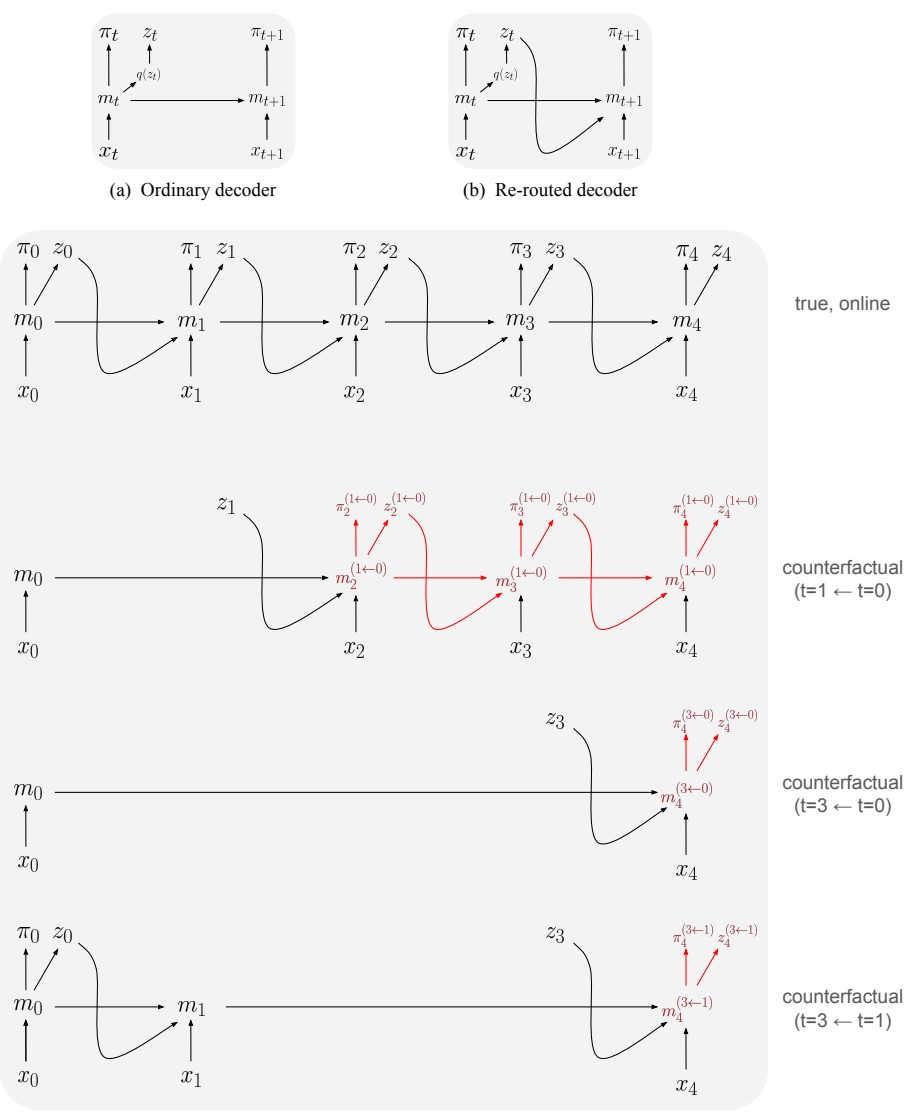

Figure 1: **Architectures. (a)** Decoders offer a common and simple solution to modeling a base system, albeit one that lacks faithfulness. **(b)** Re-routing the output of a decoder back into the base system is not sufficient to remedy this. **(c) The CST architecture** expands on the re-routed decoder by also allowing memory states to be reverted to earlier states before applying a state update. For example, in the second row, the memory state $m_1$ is reverted to $m_0$, so that the (true, online) message $z_1$ must inform the agent about changes to its current state and/or required future policy. Similarly, in the final row, the memory state $m_4$ is reverted to $m_1$, so that the (true, online) message $z_3$ must inform the agent accordingly. Black denotes variables from the true, online rollout of the agent. Red denotes variables after a counterfactual intervention. The variant of CST used (CST-RL/MR/PD) determines how the counterfactual changes are produced and trained.

**Agent construction.** We follow the standard construction of a POMDP. We denote the agent's observation at time $t$ within an episode as $x_t$, and assume that it maintains an internal state $m_t$, which we can take to be, for example, a recurrent state (such as from an LSTM), or a variable-length array of embeddings of previous observations $x_{0:t}$ (as used in a Transformer). We denote the agent's state update function as $f_\theta$, with learned parameters, $\theta$, s.t. $m_{t+1} = f_\theta(x_{t+1}, m_t)$. We denote its policy as $\pi_t = h_\theta(m_t)$.

Our construction starts with a decoder, shown in Figure 1a. A decoder maps from the state $m_t$ to an output $z_t \sim q(z_t)$, via a learned function $g_\theta : m_t \sim q(z_t)$. This mapping is trained using target data $\hat{z} \in \mathcal{D}_z$ that lives in a semantically-meaningful space. As we previously identified in Section 3, decoders benefit from being grounded, flexible, scalable, and minimally-interfering. However, their weakness is faithfulness. This is because the output $z_t$ merely reflects what information is present in $m_t$, and not whether that information is actually used in the computation of $\pi_t$.

**Defining faithfulness.** We now chart a path forward from decoders, with an aim of making them faithful. To do so, we first define two metrics for faithfulness, one weak, one strong:

**Correlational faithfulness:** if the decoder reports a value of $z_t = z$, to what degree is the behavior (from the policy $\pi$) *consistent* with this value, $z$?

**Causal faithfulness:**[4] if we were to counterfactually *intervene* on $z_t$, changing it to a value of $z'$, to what degree would this produce behavior consistent with this new value, $z'$, rather than its original value, $z$?

Causal faithfulness is a much stronger property than correlational faithfulness. It's highly desirable because it allows one to validate the agent's commitment to the value of $z_t$ at runtime by making interventions on it. This validation mechanism is especially useful when out of distribution.

**Re-routed decoders.** While it is possible for a decoder to have a degree of correlational faithfulness, it is structurally impossible for it to exhibit causal faithfulness. A simple fix is to route the output of the decoder back into the internal representation (Figure 1b), e.g. by concatenating $z_t$ with $x_{t+1}$. This amounts to incorporating the decoder output in the state update rule, with $m_{t+1} = f_\theta([x_{t+1}, z_t], m_t)$.

The difficulty with this simple solution is that there is no guarantee that $f_\theta$ will actually "listen" to $z_t$ when producing $m_{t+1}$. This is because the information in $z_t$ is largely if not completely redundant with $m_t$. This may be further exacerbated by the presence of any stochastic sampling in the generation of $z_t$ from $m_t$. Indeed, as shall be described in Section 6, our experiments find that re-routed decoders make no measurable progress beyond ordinary decoders on causal faithfulness.

How do we render the function $f_\theta$ sensitive to the value of $z$, when there is redundancy between $z_t$ and $m_t$? It is useful to frame the problem in terms of communication. The decoder pathway can be viewed as a (externally-grounded) communication channel between the internal state and itself, i.e. a form of "self-talk" or "inner speech" [67]. The problem therein is that the speaker ($m_t$) and the listener (also $m_t$) are identical, so there is no useful information in the message $z_t$ for the listener.

**Causal self-talk.** To overcome this problem, we need to create an information asymmetry between speaker and listener. We introduce ***Causal Self-Talk*** (**CST**): we encourage $f_\theta$ to be sensitive to the message $z_t$ by creating an augmented data distribution where the speaker and listener diverge. Fortunately, we have a ready source of alternate listeners: internal states $m_{t'}$ for $t' < t$.

We thus provide the decoder output with an additional role: to compactly and effectively communicate information to *earlier versions of the same agent* (from the same episode), via a semantically-grounded communication channel. Formally, this takes the form of an intervention (Fig 1c): the internal state, $m_t$, is replaced with a counterfactual one, $m_{t'}$, before being integrated with the original message ($z_t$) and next observation ($x_{t+1}$), in order to produce a counterfactual internal state, $m_{t+1}^{(t \leftarrow t')} = f_\theta([x_{t+1}, z_t], m_{t'})$. This yields a counterfactual policy, $\pi_{t+1}^{(t \leftarrow t')} = h_\theta(m_{t+1}^{(t \leftarrow t')})$.

This alone, however, is insufficient to fully specify a CST algorithm. We must also stipulate the desired outcome from enacting these interventions. We describe three alternative choices:

**CST-RL:** The messages $z_t$ should allow earlier versions of the same agent to maximize return given the current environment state. To train this, interventions must occur online (according to some schedule), and we must allow the agent to act according to the counterfactual policy following an intervention. We then use the maximization of subsequent discounted reward as the desired outcome from the intervention, via reinforcement learning (RL).

---

[4]Our use of this term is distinct from the "causal faithfulness condition" in causal modelling [64].

**CST-MR:** The messages $z_t$ should allow earlier versions of the same agent to reconstruct the next internal state, $m_{t+1}$. Unlike CST-RL, this can be done entirely in replay, without requiring the agent to ever act according to the counterfactual policy on a live environment. We add a weighted memory reconstruction (MR) term to the overall loss: $\mathcal{L}_{MR} = ||m_{t+1}^{(t\leftarrow t')} - m_{t+1}||^2$.

**CST-PD:** The messages $z_t$ should allow earlier versions of the same agent to recover the true current (and future) policy. As with CST-MR, interventions can be simulated entirely in replay. We add a weighted policy distillation (PD) term to the overall loss, using discounting function $\gamma(\cdot)$: $\mathcal{L}_{PD} = \sum_{\Delta t > 0} \gamma(\Delta t) \cdot D_{KL}\left(\pi_{t+\Delta t} || \pi_{t+\Delta t}^{(t\leftarrow t')}\right)$.

**CST and the five desiderata.** CST offers a different, less extreme tradeoff between the desiderata. By design, it inherits decoders' resolution to several desiderata. Like decoders, CST can achieve grounding by supervising the representations $z$ against a signal of known semantics. It is also equally flexible. Relative to decoders, there is some trade-off with minimal-interference as the decoder output $z_t$ is now an obligate input into the next time step. However, much like decoders, the effects on training can, in principle, be contained through appropriate gradient flow, e.g. by limiting updates to parameters of the state update function, $f_\theta$, and/or fine-tuning a pre-existing base system.

CST is highly scalable. CST-RL is straightforward to implement; in its simplest incarnation, fixing $t' = 0$ reduces to dropout on the whole internal state, $m_t$. The use of RL, however, occurs at the expense of additional data as it requires online execution of the counterfactual policy. CST-PD and CST-MR, in contrast, are entirely self-supervised: they make use of the existing trajectories generated during the training of the base system. These methods augment the data in a manner similar to the interchange intervention technique of [25], using self-imitation as the desired outcome of the intervention. Depending on the degree of non-stationarity afforded by the particular environment (and its degree of partial observability), the size of the augmented dataset can scale quadratically with the length of each episode (from choosing both $t$ and $t'$); in some settings, it may be possible to exceed this by sampling counterfactual previous states $m_{t'}$ from other episodes.

Finally, unlike decoders, CST is explicitly trained to provide causal faithfulness, by facilitating counterfactual control. Our goal in the following two sections is to evaluate how well it does this.

## 5   Experiments

**Task.** We study variants of CST in a 3D virtual environment built in Unity [1, 68]. This features a fixed-layout indoor space comprising five rooms, each with a different wall color. The agent receives as input a $1^{\text{st}}$-person visual observation and a text observation. The agent can freely navigate using a 4D continuous action space ($a \in [-1, 1]^4$) consisting of in-plane movements and a 2-DOF rotation.

We developed a simple task in this environment, called **DaxDucks** (Figure 2). We designed DaxDucks to: (1) be easy for an agent to learn a good policy; (2) require an agent to maintain and update beliefs about the latent environment state in order to maximize reward; (3) provide a source of data to ground the self-model; and (4) provide a means to evaluate the faithfulness of the learned self-model.

DaxDucks is an instruction-based, fast-binding search task, where the agent is rewarded for finding a duck that exhibits an instructed tag. Each episode consists of a sequence of trials. At the beginning of each trial, the agent's avatar spawns in the center of the middle room facing a random direction. Four identical ducks spawn at the center of the four outer rooms. Each duck is randomly assigned one of four tags ("dax", "gavagai", "kleeg", or "plork"). The agent is instructed to collect (i.e. collide with) the duck with a specified tag via the text observation channel, e.g. "Collect a gavagai." The agent may observe a duck's tag by entering the duck's room and orienting the center of its view towards the duck. This adds an additional string to the text observation, e.g. "This is a kleeg.". Colliding with *any* duck terminates the current trial, in which case the agent is respawned in the center of the middle room again and a random new instruction is delivered; a reward is also delivered if the agent collided with the correct duck. Finally, when a new trial starts, the ducks usually maintain their tags, but with a small probability ($p_{sh} = 0.1$), the tags are randomly shuffled between the ducks. Thus the agent can only maximize reward by effectively retaining a belief state over which tag belongs to which duck. Each episode lasts for 5000 steps regardless of the number of trials completed.

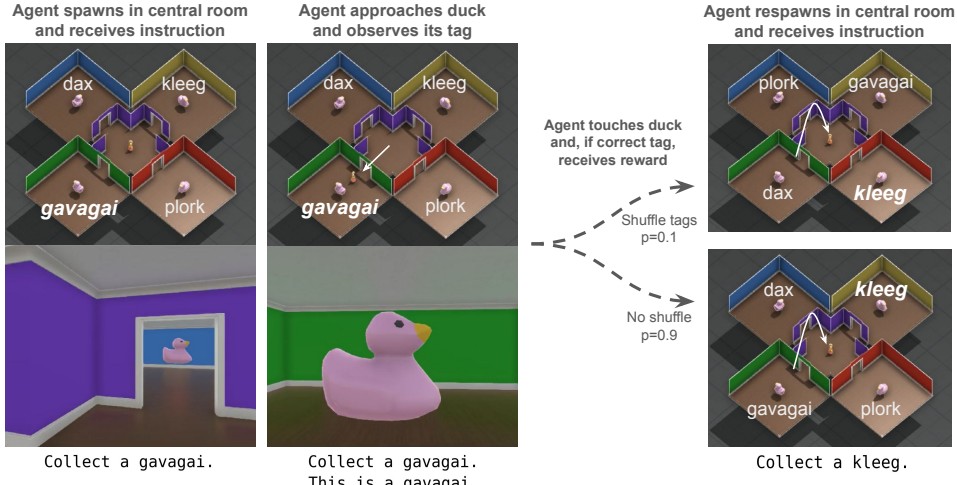

Figure 2: The **DaxDucks** task. Top-down views are for exposition only.

**Agent architecture.** For the base system, we used a standard model-free agent (Appendix A): inputs encoded with a ResNet (images) and LSTM (text); a LSTM memory module; MLPs for policy and value heads; and trained with V-trace [22]. All agents achieved a similarly high return.

We demonstrate the flexibility of grounding by considering two forms for the self-model. In the **one-hot form**, the message $z_t$ comprises four one-hot vectors (one per tag), expressing which room the agent believes each tag to be in. In the **language form**, the message $z_t$ is a synthetic language string, expressing which room the agent believes a single chosen tag to be in.

For the one-hot form (Section 6), we parameterize $q(z_t) = \prod_{\tau=1}^{4} q(z_t^\tau)$ as a product of independent categorical distributions $q(z_t^\tau)$ for the four tags $\tau$, with $g_\theta$ (an MLP) outputting the parameters of each $q(z_t^\tau)$ at time $t$, and a sample $z \sim q(z_t)$ concatenated with the encoded input at the next time step. For the language form (Section 6), we use an LSTM for $g_\theta$, sampling $z_t$ directly. Examples from a single trajectory are shown in Figure 3a (one-hot) and Figure 4a (language). We ground both forms of self-model by supervising $z_t$ against ground-truth values. For the language form, this amounts to constructing a target $\hat{z}_t$ of the form: "The `<instructed tag>` is in the `<color>` room.".

Given the simplicity of the environment, we implement all three CST algorithms using $t' = 0$ throughout. This means that CST aims to drive the decoder to communicate its beliefs to the earlier version of the agent from the start of the episode. Although this reduces the scale of data augmentation from quadratic to linear, and requires interventions on $z$ to be accompanied by a reset of the memory state to $m_0$, in return it makes it easier to measure the effects of interventions on the agent's behavior.

We used the following schedules for interventions at training time. For CST-RL, interventions $(m_t \leftarrow m_0)$ occurred randomly, with probability $p = 0.03$ that an intervention would occur at any given timestep $t$. For CST-MR, we simulated interventions in replay at every timestep. For CST-PD, we simulated interventions in replay: we divided every trajectory into a sequence of blocks of variable duration, with a $p = 0.03$ probability that a new block would start at any time $t$; we computed $\mathcal{L}_{PD}$ only up to a horizon of the end of the block, and with a constant discounting function, $\gamma(\Delta t) = 1$.

## 6 Results

**One-hot self-talk.** We start by considering the one-hot form for the messages $z$ (Figure 3a). We compare CST against two baselines: the ordinary decoder (**Ord-Dec**; Figure 1a) and the re-routed decoder (**RR-Dec**; Figure 1b). We focus on the metrics of faithfulness previously introduced.

> **Correlational faithfulness.** When no interventions are taking place, is the agent's physical behavior congruent with the output of its self-model? To measure this, we filter to all times in evaluation episodes when the agent's avatar is in the central room. We then determine the degree to which the belief attested in $q(z_t)$ matches the *next-visited* room, $r$ (whether or not

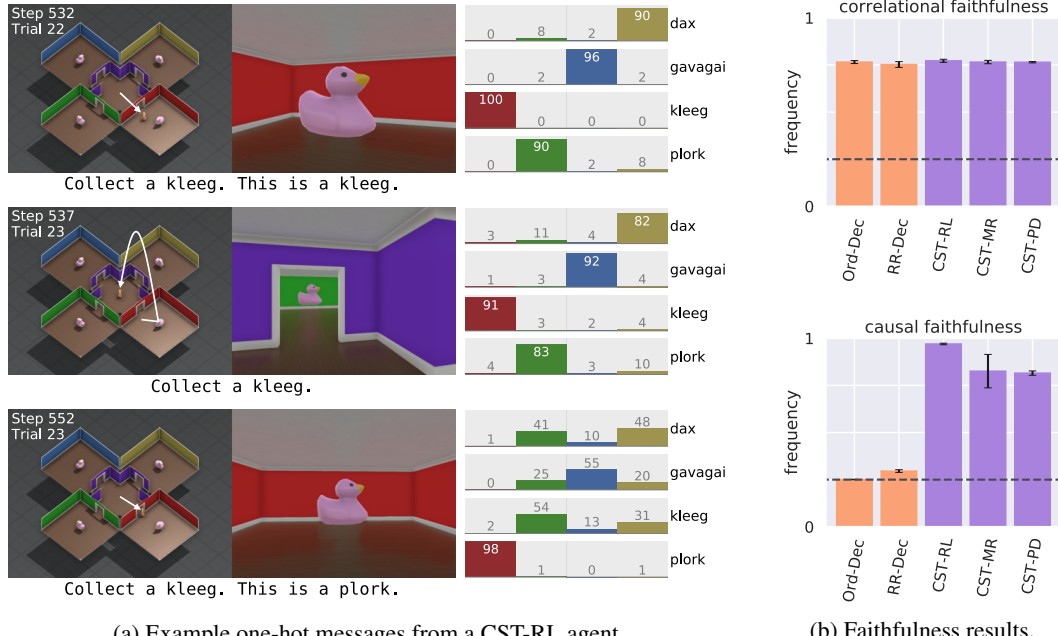

(a) Example one-hot messages from a CST-RL agent.

(b) Faithfulness results.

Figure 3: **One-hot-based self-talk. (a) Example messages during a trajectory.** Top-down view (left), agent view (middle), and $q(z_t)$ (right) during a trajectory of DaxDucks, showing the updating of $q(z_t)$ after the start of a new trial (second row) and after discovering the tags have been shuffled (third row). At each time point, the message $z_t$ is sampled from $q(z_t)$. **(b) CST renders the decoder causally faithful.** Average values across $5 \leq N \leq 10$ training runs per method; error bars show 95% confidence intervals. Dashed lines indicate faithfulness expected from random messages.

the instructed tag is actually in $r$). For the one-hot message form, this amounts to reading off the value of $q(z_t^\tau = r)$ for the instructed tag $\tau$.

**Causal faithfulness.**    When we intervene on $z_t$, is the agent's subsequent physical behavior congruent with this new value? To measure this, we run evaluation episodes where, at the start of each trial, we inject a message $z'$ indicating that the instructed tag is in some random room $r'$. We then measure the probability that the agent actually visits room $r'$ next.

CST maintains comparable levels of correlational faithfulness as the baseline decoders. However, the decoder baselines failed to exhibit a meaningful degree of causal faithfulness. This is indeed impossible for **Ord-Dec** given the lack of any causal pathway from $z_t$ to $m_{t+1}$. Re-routing $z$ back to the internal state in **RR-Dec** does result in a small increase above chance, but the effect is relatively tiny. In contrast, all CST objectives substantially increase the degree of causal faithfulness (Figure 3b).

**Language-based self-talk.**    Next, we trained agents to model their belief state using messages $z$ in a synthetic language form (e.g. Figure 4a). For correlational faithfulness, we obtain analogous values of $q(z_t^\tau = r)$ by computing the likelihoods of generating the four messages "The <instructed tag> is in the <color> room.", and normalizing these to sum to 1.

As with the one-hot form, we found that the greatest gain of CST was its ability to imbue the self-model with causal faithfulness. The effect was most profound with CST-RL, and moderate with CST-PD. However, we found it difficult to get any traction with CST-MR; this was either disruptive to learning the base policy, or ineffective at causal faithfulness.

We note that the faithfulness results with the language form were overall weaker than for the one-hot form. This may be because the one-hot messages encode a room belief about all four tags.

**Semantic control.**    As a final test, we consider whether CST endows the self-model with sufficient causal power to act as an effective semantic control interface for the agent. We thus measure how

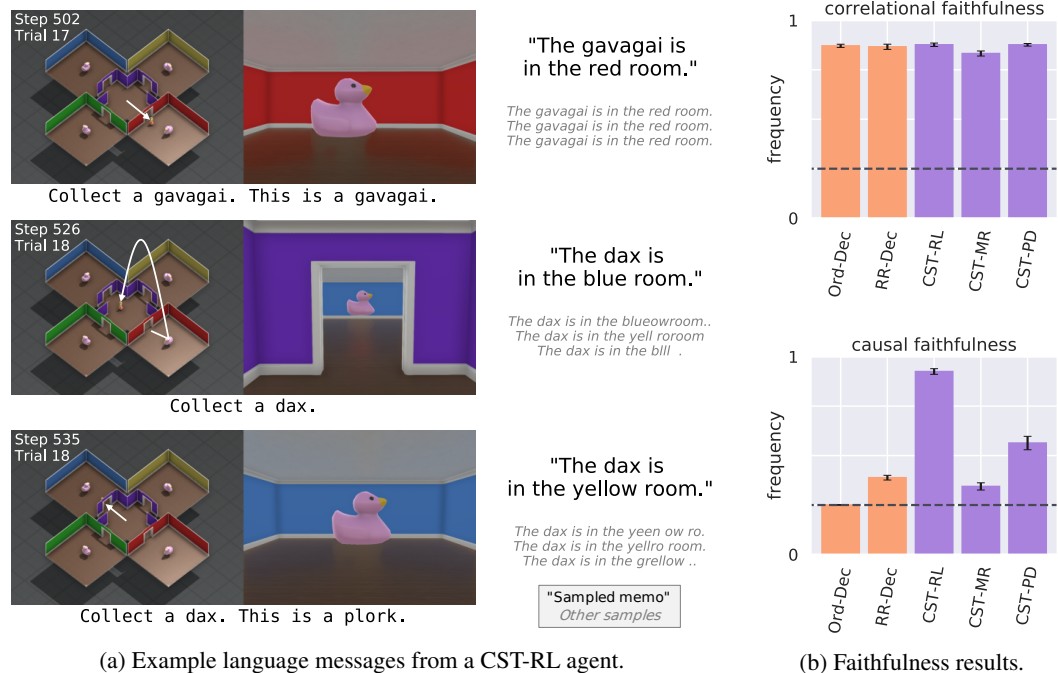

(a) Example language messages from a CST-RL agent.

(b) Faithfulness results.

Figure 4: **Language-based self-talk.** Results as in Figure 3.

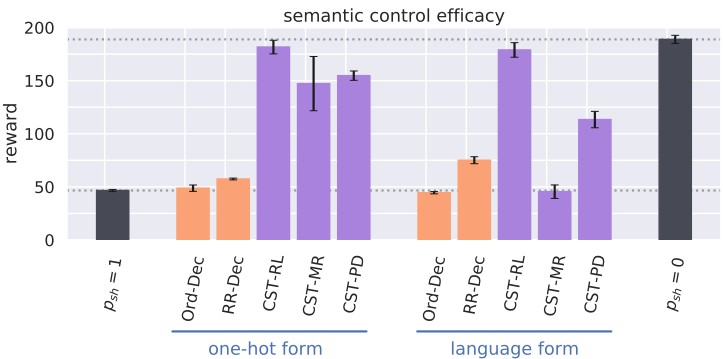

Figure 5: **Efficacy of semantic control via interventions on the self-model.**

effectively the agent can use injected information to obtain reward. We start with two baselines: (1) the return on evaluation episodes where $p_{sh} = 1$, i.e. where tags are shuffled on every trial; and (2) the return on evaluation episodes where $p_{sh} = 0$, i.e. where tags are never shuffled. Respectively, these represent the bounds of rewards the agent naturally obtains when it can acquire (1) minimal or (2) maximal information about the location of the instructed tag. We compare these to the reward obtained on evaluation episodes where information about the tag location is provided through intervening on $z$. Specifically, we set $p_{sh} = 1$, and inject a message $z'$ at the start of each trial indicating the location of all the tags (for the one-hot form), or the location of the instructed tag (for the language form). The results (Figure 5) show that CST not only furnishes an agent with a semantically-grounded model that describes its behavior, but also one which can be used as a mechanism to guide it.

## 7 Discussion

We articulated five key desiderata of an ideal explanation system for an AI system: grounding, flexibility, minimal-interference, scalability, and faithfulness. Unlike existing approaches, we seek to satisfy all of these by training the base system to serve a causal model of itself. We developed an instance of this solution for Deep RL agents: Causal Self-Talk. In CST, an agent learns to serve a

grounded and faithful model of its internal state by learning to communicate with itself across time. We demonstrated its effectiveness under various forms in a simple embodied task in a simulated 3D environment. We also showed that CST enables a new mechanism for effective semantic control.

We presented three variants of the CST algorithm. CST-RL yielded the greatest degree of causal faithfulness and efficacy of semantic control. Here, online interventions at training time force the agent to learn how to use the content of its self-model to maximize its reward. CST-RL's success, however, may reflect features of the particular task setting we study: here, there is a close alignment between writing a faithful $z$ and maximizing reward. This may not always be the case.

In contrast, CST-PD offers an attractive alternative for two reasons: (1) it trains entirely in replay, and thus potentially interferes less with the base system; and (2) as it primarily seeks to enable *imitation* of the current behavior, the faithfulness of the self-model is targeted by design. One challenge here is learning to capture the important features of behavior; more impressive gains may be possible using richer imitation learning techniques [e.g. 34] rather than behavioral cloning. We had the least success with CST-MR: there also may be theoretical obstacles to deploying CST-MR in some settings, e.g. when the dimensionality of the internal state is very large, or is of variable length.

**Limitations.** In our experiments, we use a limited form of CST, where we exclusively revert memory states $m_t$ to the state at time $t' = 0$, rather than the more general case of $t' < t$. We do this to simplify the presentation and evaluation of the technique, but it imparts two limitations: (1) it requires $m_t$ to be reverted to $m_0$ in order to effectively intervene on the message $z$; and (2) it requires $z$ to contain all behaviorally-relevant information. While sufficient for the simple task at hand, this becomes unrealistic in more complex environments. When $z$ contains only a partial description of $m_t$, it will be necessary for the agent to be able to re-integrate injected $z$ values with its current memory state. The more general case of $t' < t$ offers a viable path to achieving this.

Utility requires that the self-model be grounded in a semantically-meaningful form. We have shown how a ready source of data can be used to train the self-model. How this method stands up in more limited data regimes, or in OOD scenarios where the self-model hasn't been supervised, remains an open question. It may be possible in some domains to leverage semi-supervised methods, and/or to impose structure on the self-model as a helpful constraint.

**Future opportunities.** We focused here on expressing the agent's beliefs about the current state of the environment in the self-model. A complete answer to the question, "Why did you do that?" might also include an expression of an agent's goals, plans, and model of the world [53]. The schema we have developed here should be sufficiently flexible to extend in this direction, given the right training data. It would also be of great value to determine how to apply these techniques beyond RL agents.

Our primary focus in this paper has been on building self-models to yield explanations. However, this is only one of several utilities that a faithful self-model might deliver. For example, we demonstrate that causal self-models provide a semantic control mechanism. This could be useful for guiding agent learning through human interaction [1], metacognitive reporting and control [9, 24, 29], or as an interface for safety-critical applications [41, 45, 51, 61]. By learning to express its internal state through a different representational schema, this may also open up avenues for new exploration or hierarchical reinforcement learning techniques.

Finally, training agents to constantly talk to themselves might prove to be a data-efficient way of enabling agents to talk to others (and us!). Ultimately, this may pave a way for us to invert of the flow of knowledge, allowing agents to not only answer, "Why did you do that?", but also "How?".

## Acknowledgments and Disclosure of Funding

The authors would like to thank Edgar Duéñez-Guzmán for early environment design, Hamza Merzic for agent infrastructure assistance, Allison Tam for help in agent development, Chen Yan for useful discussion, Lucy Campbell-Gillingham for organizational support, and Johannes Welbl, Kory Mathewson, Pedro Ortega, Agnieszka Grabska-Barwińska, Chen Yan, Matt Botvinick, and Lisa Anne Hendricks for comments on the work and/or the manuscript.

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
