# A Agent architecture and hyperparameters

We note that our emphasis in this work was not on finding the overall best performing networks, so we did not extensively tune network and learning hyperparameters.

We trained agents using a distributed RL setup, with 4096 parallel actors. We trained the one-hot form using a 4x4 TPUv2, and the language form using a 4x4 TPUv3. Training runs took approximately 12-48 hours to reach maximum episode return ($\sim$ 200/episode), typically after 50-200k learner steps.

| | | | |
|---|---|---|---|
| | | Input resolution | $(160, 192, 3)$ |
| | Image encoder $e_\theta^i$ | ResNet | number of blocks — 3
channels per block — $(16, 32, 32)$
conv layers per block — $(2, 2, 2)$
conv filter size — 3
nonlinearity — ReLU
max-pool filter size — 3
max-pool strides — 2 |
| State update $f_\theta$ | String encoder $e_\theta^s$ | Tokenizer | tokenizer name — subword
vocabulary size — 8000
max token length — 19 (right-padded) |
| | | Linear embedding | embeddings per token — 16 |
| | | LSTM | hidden units — 256 |
| | Memory core | Input structure | $[e_\theta^i(i_t), e_\theta^s(s_t), a_{t-1}, r_{t-1}]$ |
| | | LSTM | hidden units — 512 |
| Policy head $h_\theta$ | | Policy MLP | hidden units — 200
action space — $\in [-1, 1]^4$ |
| | | Value MLP | hidden units — 200 |
| CST head $g_\theta$ | | MLP | hidden units — 32 (one-hot)
512 (language) |

Table 1: Agent architecture.

| | | |
|---|---|---|
| V-Trace Loss | baseline cost | 1.0 |
| | entropy cost | 0.001 |
| | $\gamma$ | 0.95 |
| | max reward | 1.0 |
| Adam Optimizer | learning rate | $1e^{-4}$ |
| | $\beta_1$ | 0.0 |
| | $\beta_2$ | 0.95 |
| | clip grad norm above | 40 |
| Schedule | batch size | 192 |
| | termination steps | $6e^7$ |

Table 2: Training hyperparameters.