# OpenReview forum: "Explainability Via Causal Self-Talk"
_NeurIPS.cc/2022/Conference — NeurIPS 2022 Accept_

### Official Review · Reviewer_g2gH · 2022-07-01

**Rating:** 9
**Confidence:** 3
**Soundness:** 4 excellent
**Presentation:** 4 excellent
**Contribution:** 3 good

**Summary:**

In this paper, the authors define 5 desiderata for AI explainability: grounding, flexibility, minimal-interference, scalability, and faithfulness. They show how these desiderata are satisfied by a system that learns an internal model via "causal self-talk" (a data augmentation strategy in which externally grounded information is decoded and then rerouted back into the internal state). Intuitively, the decoded information functions as an explanation for why the system behaved in a particular way. To enforce "causal faithfulness" (the decoded information exerts a causal influence on the system's output), the authors train the decoder to communicate information to earlier versions of the same agent. This creates the information asymmetry necessary for the agent to use the decoder output. The authors explore 3 different objective functions for training the decoder (based on reinforcement learning, memory reconstruction, and policy distillation). They find that the reinforcement learning variant performed best in terms of causal faithfulness and "semantic control" (using decoded information to obtain reward).

**Questions:**

If the authors can address the two main weaknesses raised above (somewhat contrived experiments and lack of comparison to other approaches) then I will be happy to increase my rating.

UPDATE: the authors have addressed these concerns. Although nothing really substantive has changed in the paper, I understand their reasoning as well as the page constraints.

**Limitations:**

I don't see any major negative societal impact of this work. I thought the authors did a good drop addressing limitations and future directions.

**Strengths And Weaknesses:**

Strengths:
- The work is original, as far as I can tell (I am not an expert in this area).
- The paper is clearly written and logically argued.
- The ideas are interesting.
- The experiments demonstrate the efficacy of the proposed method.
- Given that explainability is an important real-world problem, the results are potentially significant.

Weaknesses:
- I thought the experiments were useful for demonstrating the efficacy, but they also felt rather contrived relative to real-world problems where explainability is needed.
- Only variations of the proposed system were studied. This has the advantage of provided well-controlled comparisons, but it doesn't establish to what extent the system outperforms previously proposed systems.
- This is perhaps less important, but I would have thought that human judgment of explanatory adequacy is important for evaluating real-world explanation systems. This matters at the end of the day, right?

---

> ### Author Response · Authors · 2022-08-02
> **Response to Reviewer g2gH**
>
> We thank the reviewer for their comprehensive and thoughtful review. We are especially grateful for the characterization of our work as original, clear, and significant. We also appreciate providing a clear and accurate summary of our work – we couldn’t have done this better!
>
> > **I thought the experiments were useful for demonstrating the efficacy, but they also felt rather contrived relative to real-world problems where explainability is needed.**
>
> The goal of this paper is to clearly articulate the nature of the problem, present a viable path forwards, and to show how it works. As the reviewer indicates, the experiments were targeted directly at these problems, and we are glad that the reviewer appreciated their success in achieving this.
>
> Applying our technique in more complex and real-world environments is definitely a necessary future step, but there is unfortunately already too much to be done within 9 pages. Our environment was chosen to facilitate a clean demonstration that our algorithms are doing the right thing. We hope that the solid foundation of results made in this work convinces the community to extend our techniques to real-world domains.
>
> > **Only variations of the proposed system were studied. This has the advantage of provided well-controlled comparisons, but it doesn't establish to what extent the system outperforms previously proposed systems.**
>
> In developing this work, we undertook extensive literature review, which we summarize in Sections 2-3. Our conclusion was that existing XAI techniques fail on the criteria we care about _a priori_. To the best of our knowledge, there aren’t any existing techniques to compare against.
>
> The only comparison we really were able to do were against ordinary decoders (and Ord-Dec) and re-routed decoders (ReR-Dec). We show that CST outperforms these in causal faithfulness.
>
> We sincerely hope and expect CST to be outperformed in future.
>
> > **I would have thought that human judgement of explanatory adequacy is important for evaluating real-world explanation systems.**
>
> We agree! And we are buoyed by recent empirical works (e.g. [Abramson et al. (2022)](https://arxiv.org/abs/2205.13274); [Thoppilan et al (2022)](https://arxiv.org/abs/2201.08239); [Kiela et al, (2021)](https://arxiv.org/abs/2104.14337)) that introduce techniques for incorporating human judgements in benchmarking an agent’s or a language model’s performance, as well as recent sociological and psychological perspectives on this question (e.g. [Miller (2017)](https://arxiv.org/abs/1706.07269) and [Miller (2020)](https://arxiv.org/abs/1811.03163)).
>
> Nonetheless, we argue that there is a set of technical requirements for an explanation system that must be satisfied before human judgment can even come into play. The focus of our work in this paper is on addressing these.
>
> ---
>
> We appreciate that the reviewer gave actionable guidance as to how we could convince them to raise their rating. We hope that these responses adequately address the questions specified by the reviewer; if so, we ask the reviewer to consider increasing their rating as suggested.

---

> > ### Comment · Reviewer_g2gH · 2022-08-07
> > **increasing my rating**
> >
> > Thank you for the response to my comments. As promised, I will increase my rating to 9.

---

### Official Review · Reviewer_1Udf · 2022-07-09

**Rating:** 5
**Confidence:** 3
**Soundness:** 1 poor
**Presentation:** 2 fair
**Contribution:** 2 fair

**Summary:**

The authors delineate a set of desiderata for interpretabiliy of RL systems and then build a causally inspired interpretable solution in which the agent generate a textual output to describe its internal state and utilizes it as a form of memory to update its internal state during the course of a trajectory rollout. The causal aspect stems from interventions on the internal state of the agent being set to its initialization value. The authors evaluate their method on their own benchmark.

**Questions:**

1. Do you think the intervention could hamper policy learning in the complex environments? It appears that “resetting” the internal state constantly to the initialization value could potentially be harmful.
2. Do you have any plans for releasing code + environment?
3. Have you considered how the method might be used for language-conditioned imitation learning? One could feed in instructions instead of z from the previous time step. This could work like a conversational RL agent where it takes an instruction and performs some actions and then returns a string of text for conversation.


**Limitations:**

I believe the paper in its current form needs significant improvement. There seems to be a disconnect between the evaluation and the desiderata which can be bridged through thorough experiments. Additional work is necessary before the work is ready for publication. I wish the authors all the best for this.

**Strengths And Weaknesses:**

Strengths:
1. Well Written paper: The authors describe not only the intuition behind their work but also how the intuition came into being. This is useful when learning to do research in a new field.
2. Important problem: The authors have selected an important problem. Explainability of RL agents is key to making RL agents safer and interpretable.
3. New evaluation environment. The authors have built their own RL environment which (if open-sourced) could yield future results.
4. Detailed Desiderata: The authors define their own set of desiderata for interpretable RL systems.

Weaknesses:
1. No pictorial representation of architecture. The authors should provide a figure depicting a rollout of the RL agent in the environment. I can conceive of this by representing an MDP along with a rollout of the policy for a few steps.
2. Biggest Gains are in the most simple settings: Providing the textual input as a one-hot encoded vector generates most gains. The authors could try to pretrained embeddings for text to improve this.
3. Highly simplistic evaluation setting: There exist multiple environments with textual annotations, e.g., https://openreview.net/forum?id=eCPCn25gat which could allow for similar experiments. Experiments on them would be useful. My worry is that the approach will not scale.
4. Not enough explanation delineating the relationship between experiments and desiderata: While defining desiderata is helpful, a callback to the desiderata is necessary in the experiments sections describing how they were achieved. This is missing.
5. Missing citing work: https://arxiv.org/abs/2010.03110 and https://proceedings.neurips.cc/paper/2021/file/c1722a7941d61aad6e651a35b65a9c3e-Paper.pdf and https://ieeexplore.ieee.org/document/9561439

---

> ### Author Response · Authors · 2022-08-02
> **Response to Reviewer 1Udf: 1/2**
>
> We thank the reviewer for their thoughtful review. We are especially grateful for the recognition of our work as well-structured and important.
>
> > **There seems to be a disconnect between the evaluation and the desiderata which can be bridged through thorough experiments.**
>
> We take this to be the headline of the reviewer’s critique, and that this, in turn, breaks down into two major pieces:
>
> * **Weakness #4: We don’t sufficiently explain the relationship between the experiments and the desiderata.**
> * **Weaknesses #2 & 3: The experiments themselves are too simplistic to show that the desiderata are satisfied.**
>
> Regarding Weakness #4, we recognize that we did not spell this out clearly enough in the initial submission of the paper. Nonetheless, this is easily addressed:
>
> * Because the CST algorithm develops upon decoders (which are standard architectural components in Deep RL), CST inherits decoders’ properties of groundedness, flexibility, and scalability _by design_, and thus these properties do not require experimental validation. We do demonstrate two forms for the self-model (one-hot and language), but this is merely to illustrate flexibility and grounding, not to experimentally validate that these are satisfied.
> * CST also largely inherits the properties of minimal interference from decoders _by design_. Like decoders, one can tune the degree to which CST alters the weights of the base system (for better or for worse) by choosing how to propagate the additional gradients. The only experimental test required is validating that the introduced changes do not disrupt policy learning. This indeed was our finding.
> * Faithfulness is the major desideratum that requires experimental testing. This is the focus of our experiments in Sections 5 and 6.
>
> We have clarified this by changing the text at the end of Section 4, under the heading “CST and the five desiderata”.
>
> Regarding Weaknesses #2 & 3, we appreciate the reviewer’s point, and agree that there is much to do to push this technique further toward richer forms of self-model and richer environments. From what we have seen so far, we are confident that the method will scale to larger problems.
>
> However, we argue that a first paper on this approach should not be about scaling up. It is first necessary to clearly articulate the nature of the problem, to present a viable path forwards, and to demonstrate how and why it works. We explicitly designed an evaluation setting that would foreground the problems at hand, and allow us to carefully measure whether the algorithms are doing the right thing. We ask the reviewer to consider that the experiments themselves are pitched at just the right level to evaluate the fundamentals: that CST satisfies the desiderata at hand.
>
> We agree that there is much to do to push on scaling up and performance. Nonetheless, trying to cram more into the already-packed 9 pages would be at the significant detriment of clarity, readability, and accessibility (it is already pushing it to the limit!). We believe that the paper as it is covers the appropriate scope in establishing the fundamentals. We hope that the reviewer will give us the opportunity to publish this, so that we and the community can steer our focus to exactly these concerns.
>
> > **Weakness 1: No pictorial representation of architecture.**
>
> To assist in exposition, we have now: (1) amended the pictorial representation in Figure 1 to clarify that `z_t` is fed back into the agent on the next time step, and (2) included an expanded version on this figure in the supplement, as Figure S1.
>
> > **Weakness 5: Missing citing work.**
>
> We thank the reviewer for these suggestions. We looked into them, but do not think they are ultimately directly related. These studies involve building an agent’s knowledge of the causal structure of the environment, either through curiosity about causal relationships in MDPs ([Sontakke et al (2020)](https://arxiv.org/abs/2010.03110)), measuring how much causal influence one’s actions have at any time ([Seitzer et al (2021)](https://proceedings.neurips.cc/paper/2021/file/c1722a7941d61aad6e651a35b65a9c3e-Paper.pdf)), or constructing state abstractions that foreground the causally relevant variables in environment dynamics ([Lee et al (2021)](https://ieeexplore.ieee.org/document/9561439)). In contrast, CST does not aim to build casual knowledge about the environment. Rather than learning about some causal relationships in the extrinsic environment, CST aims to induce causal relationships inside of the agent itself (i.e. make the actions not just correlated with the messages `z_t`, but causally dependent on them).

---

> > ### Comment · Reviewer_1Udf · 2022-08-09
> > **Thanks for your rebuttal**
> >
> > Thank you for your response. I think the first step in algorithmic development is understanding how an approach scales to varying problem complexities. Having said that, I do agree that there is some novelty in the approach and that there is some benefit in publishing the approach in its current form. For this reason I am happy to increase my score.

---

> ### Author Response · Authors · 2022-08-02
> **Response to Reviewer 1Udf: 2/2**
>
> _(note that this is part 2/2)_
>
> > **Q1: Do you think the intervention could hamper policy learning in complex environments?**
>
> The reviewer is correct that online resetting interventions could indeed have such detrimental effects. There are two straightforward solutions to this.
>
> First, for simplicity, we focus our results on resetting interventions (i.e. where the memory state is reverted to the state at time `t’ = 0`). However, this is just the simplest case, and one can also intervene by “rewinding” memory back to some earlier time (`t’ > 0`). This is a less drastic intervention. We have done some experiments with this more general case, and it indeed works. We haven’t included any results because the paper is already packed enough as it is; also, in the simpler environment we consider, there’s no additional benefit in using the more general method.
>
> Second, and perhaps more importantly, if there’s a concern about such interference, one should consider using CST-PD rather than CST-RL. In the RL variant, interventions are executed online, which as the reviewer intuits, can disrupt the data distribution on which the policy is trained. The PD variant, in contrast, operates entirely in replay, so disruption to policy learning is at most indirect, and far less likely. Moreover, as discussed in the main text, it is possible to contain the gradient flow from the CST-PD objective to mitigate or prevent interference with policy learning altogether.
>
> > **Q2: Do you have any plans for releasing code + environment?**
>
> We would very much like to release both the code and the environment, and are actively looking into making this possible. We are currently blocked while waiting for an upstream library dependency to be open-sourced.
>
> > **Q3: Have you considered how the method might be used for language-conditioned imitation learning?**
>
> An excellent question, and very similar to one raised by Reviewer KW43. We address the topic at length in our response to their review, which we hope this reviewer finds satisfying.
>
> ---
>
> We hope that each of the reviewer’s concerns have been addressed. If so, we ask that the reviewer reconsider their recommendation that this paper be rejected.

---

### Official Review · Reviewer_KW43 · 2022-07-11

**Rating:** 6
**Confidence:** 3
**Soundness:** 3 good
**Presentation:** 3 good
**Contribution:** 3 good

**Summary:**

This paper focuses on the problem of improving the faithfulness and explainability of AI system. The authors proposed a architecture based on POMDP and add a decoder to the internal representation to generate semantically meaningful information that could be further leveraged for internal state update and future policies. The authors showed that by intervening on the semantically grounded information, models could learn from the causal self-talk by incorporating prior internal representations with new semantic information. The experimental results show improvement of the models causal faithfulness (i.e., when passed in new semantic information, how often will the model act following the guidance) in a synthetic environment.

**Questions:**

As stated previously, my questions are as follows:
1. How significant is the self-talk process compared to grounded representation learning that requires $z_t$ and $m_t$ to be aligned?
2. The proposed 3 causal-self talk models have similar performance under current experimental settings. I was wondering if there will be specific scenarios where some of them would be best (e.g., is CST-MR always subordinate compared to the other two or could it be useful in some specific scenario). It will also be interesting if the authors could discuss if the composition of the three methods is feasible and will perform well (since each of them works toward the same goal)


**Limitations:**

The authors have stated their limitations thoroughly in the paper.

**Strengths And Weaknesses:**

[+] In this paper the authors listed five desiderata for explainable AI systems and designed a corresponding model architecture that contains semantic outputs at each time-step. The proposed architecture is general enough to be a common backbone of explanatory models with different design choices of intermediate representation used and decoder architectures.

[+] The proposed causal self-talk asks the model to condition on the semantic output $z_t$ of itself as input and do counterfactual reasoning given a state $m_t'$ that is not aligned to the $z_t$. With this intuitive design, the model can learn to ground its internal representation and policy on the semantically meaningful representation. This makes the model follows from the explanations $z_t$ it provided and could be acting faithfully with new semantic guidance.

[-] One concern of this paper is on the causal faithfulness. The current definition provided by the authors is to what degree would the model act consistently with the intervention of semantic input $z_t$. This definition to me seems directly related to some current studies in language-conditioned policies learning or grounded policy learning. As the overall goal is to have model that not only acts with its own semantic output but also act consistently given new semantic output, there should be other offline learning methods that solves similar issue. This makes potentially makes the self-talk process less significant.

---

> ### Author Response · Authors · 2022-08-02
> **Response to Reviewer KW43**
>
> We thank the reviewer for their thorough review. We especially appreciate the highlighting of our core contributions as the major strengths of the paper.
>
> > **Relationship to grounded representation learning**
>
> We understand the reviewer’s primary concern to be that CST may bear a relationship to existing techniques in language-conditioned and grounded policy learning, which may impact the relative significance of our contribution.
>
> Indeed there is a relationship! – but also two major points of difference.
>
> (1) While both our work and this other literature seek to ground part of an agent’s representations using pre-existing structures, the goals are very different.
>
> In the grounded representation learning literature, the goal is to scaffold agent representations using these pre-existing structures (typically language) in order to yield better-performing policies and more systematic generalization. This can take the form of grounding the agents with language inputs (e.g. [Hermann et al (2017)](https://arxiv.org/pdf/1706.06551.pdf), [Hill et al (2020)](https://arxiv.org/pdf/2009.01719.pdf)), language outputs (e.g. [Zellers et al (2022)](https://arxiv.org/pdf/2106.00188.pdf), [Lampinen et al (2022)](https://proceedings.mlr.press/v162/lampinen22a/lampinen22a.pdf)), or structured intermediate representations (e.g. [Koh et al (2020)](http://proceedings.mlr.press/v119/koh20a/koh20a.pdf)).
>
> In contrast, the primary goal of our paper is not to improve policy performance or systematic generalization, but rather to build an explanatory model of the agent. With this goal, the grounding of the model’s representation is a basic necessity (otherwise external users can’t use it), but it is by no means sufficient for success. In fact, in the domain we consider, grounding is one of the easier problems to solve, and we largely take it for granted in the paper. The far more interesting and novel contributions of our paper are with respect to the other desiderata, particularly faithfulness, which are not issues generally addressed by this other literature.
>
> (2) While both our work and this other literature consider “semantic control” interfaces to agents, these take on very different forms and play very different roles in the respective research.
>
> In the grounded representation learning literature, semantic control is often a primary objective in and of itself: researchers want to equip agents with language inputs so that users can issue instructions in a familiar medium. The agents must thus consume these instructions as _inputs_.
>
> In contrast, in our work, semantic control is not a goal in itself, but rather a means for validating the faithfulness of the agent’s outputs. Rather than explicitly training the agent to follow instructions as an _input signal_, we are training the agent to make a strong enough commitment to its _outputs_ such that if they were counterfactually changed, they could function as a control mechanism. Thus the site of control is different (inputs vs outputs) as well as when control occurs (at training time vs at evaluation time only in our case).
>
> (Finally, the fact that this control mechanism is “semantic” is really just a bonus from having grounded the outputs; the interesting achievement here is that counterfactual interventions on an agent’s _output_ can be made into a control mechanism in the first place. This is what CST enables.)
>
> > **It will also be interesting if the authors could discuss if the composition of the three methods is feasible and will perform well.**
>
> The reviewer raises an interesting question. Our intuition is that the composition of the three methods is feasible, and might be desirable in some domains, as the different objectives have distinct strengths and weaknesses.
>
> For example, CST-RL delivered the best performance in the data regime we explored, but this comes at the requirement of external reward signals and online learning. CST-MR generates dense learning signals while also permitting offline learning, but may be prone to overfitting as the agent attempts to reconstruct the entirety of its internal state. It is also less appropriate for high-dimensional agent representations, such as transformers. CST-PD strikes an interesting balance between the two, as it privileges reconstructing those features in the internal representation that actually have behavioral consequences.  One can also imagine other variants, that attempt to distill value functions, or generalized value functions, or that use contrastive objectives or learned discriminators.
>
> These are interesting ideas to speculate upon. For the moment, we believe it is best to focus the paper on simple and straightforward demonstrations of the methods in isolation due to space constraints.
>
> ---
>
> We hope that we have resolved the reviewer’s only stated concern with the paper. If so, we ask that the reviewer consider raising their score accordingly.

---

> > ### Comment · Reviewer_KW43 · 2022-08-09
> > **Discussion**
> >
> > Thanks the authors for their clarifications. I will keep my original ratings.

---

### Official Review · Reviewer_AH1b · 2022-07-22

**Rating:** 5
**Confidence:** 2
**Soundness:** 2 fair
**Presentation:** 3 good
**Contribution:** 3 good

**Summary:**

This paper introduces a new approach to explainable models called CST, or causal self-talk. There are 4 variations presented. The paper also defines several “desiderata” that should guide explainable AI systems such as faithfulness and scalability. Authors claim only self-models can maximize desiderata and CST is an example of that. There are experiments of CST-based models in a virtual environment called DaxDucks.

**Questions:**

Questions:
1.Pg. 1, line 30 “ We argue that only the AI system itself is capable of supplying a model that fulfills all the key desiderata”: this seems to be a big claim. I’m not convinced this is argued considerably in the paper or really the focus. The paper does propose one such method but doesn’t build a case for it being the only such system.
2.Section 2: I would elaborate this section more. Especially “faithful” since that seems to be multiple desiderata in and of itself and vague. Also maybe add in-line citations.
3.Does this method only work for single time-step instances?

Typos and Writing suggestions:
1.Pg. 3, line 86, “au naturale”: odd word choice
2.pg . 1 line 58, “to apply it to any network” change to “to apply it to any size network”
3.Pg. 1, line 30 “key desiderata”: I would flesh out these desiderata (which are discussed in section 2.
4.Pg. 3, line 117 “feed the model’s outputs as inputs”: this seems to be the crux of the whole paper. I would introduce this way earlier.
5.Pg.2 line 59 “accurate, factual, and truthful”: seems to all mean the same thing?


**Limitations:**

I can tell this paper was a lot of work to write. There’s lots of parts (good) but they're all kind of underdeveloped (bad). The main limitation is that it’s very hard to follow. The methods and experiments sections can be clarified. If the authors can do that successfully, this could be a huge improvement. Also, the paper tries to do a lot of things in just 9 pages. There’s the desiderata definitions and start off sounding like it’s a conceptual/theoretical paper. Then it introduces and experiments with four different variations of CST.

**Strengths And Weaknesses:**

Originality:
This paper is original in introducing and testing out the new concept of causal self-talk for explainable AI. It gives a pretty thorough coverage (concepts, methods, and experiments) of the new idea and offers several variations. The paper also defines concepts good to optimize in explainable AI.

Quality:
This paper has mixed quality. Some sections look underdeveloped. Section 2, for instance, seems pretty important considering their claimed contribution but looks not fully fleshed out. I can tell the authors did a lot of work for this paper and perhaps there’s much valuable insights here but it’s lost in the presentation and explanation.

Clarity:
The paper is well written and clear in some sections (lit review and historical overview) but not others (methods and experiment). The writing is somewhat experimental/creative(?) in some parts. Section 4 in particular, is confusing. Maybe Figure 1 can be improved to better relay the idea of time-step inputs?


Significance:
Their newly introduced concept is interesting and paves a path forward for the XAI community on causal methods. Their limitation of memory states not currently workable for general multi-time steps is a big one. Overall it’s an interesting early idea and implementation that can be built on with future work.

---

> ### Author Response · Authors · 2022-08-02
> **Response to Reviewer AH1b**
>
> We thank the reviewer for their careful review and thoughtful comments. We especially appreciate the recognition of our work as thorough and original.
>
> > **There’s lots of parts (good) but they're all kind of underdeveloped (bad). The main limitation is that it’s very hard to follow.**
>
> First, we feel it is important to address the reviewer’s concerns regarding presentation, especially with respect to the density of the paper, its overall structure, and the rationale behind writing it in this particular way. We acknowledge that there’s a fair amount of ground to cover!
>
> We believe that to be impactful, this paper must achieve two goals:
>
> * (A) Identify conceptual issues in the XAI field, and iron them out (Sections 1-3)
> * (B) Once this is in place, identify novel solutions to the XAI challenge (Sections 4-6)
>
> We believe that (A) is an important contribution, but not enough for a technical paper; however, (B) cannot be done unless (A) is already in place. We agree (A) and (B) together make for a dense 9 pages, but limiting it to either (A) or (B) would make for a substandard paper.
>
> Nonetheless, we recognize that the reviewer’s concerns about clarity are valid, and so have taken several steps to make the presentation more digestible to the reader:
>
> * Updated the final paragraph of the introduction to better prepare the reader for the transition from theory in Sections 1-3 to practice in Sections 4-6.
> * Improved Figure 1, as suggested, so that the message `z_t` is more clearly depicted as an input to the network.
> * Added an additional supplemental Figure S1 that illustrates the CST approach in more detail.
> * Updated Section 2 to explicitly flag that the desiderata definitions are more fleshed out in Section 3.
> * Added more subsection headings in Section 4.
> * Reduced the (admittedly!) “creative” writing throughout the paper, especially at the beginning of Section 4.
>
> ### Other comments:
>
> > **Q3) Does this method only work for single time-step instances?**
>
> > **Their limitation of memory states not currently workable for general multi-time steps is a big one.**
>
> This is not actually a limitation of the method. We simply chose to focus on always reverting to `t’=0` in our presentation in order to simplify our (already complex) paper and clarify our evaluation metrics. Indeed, we have experimented with the “multi-time step version” of the technique (where `t’` could take on any value), and it indeed works, though it doesn’t offer any particular advantage in the evaluation setting we present. Exploring the multi-time step version would be more interesting and useful in more complex domains, and we have updated the Discussion to make clear that this is a valuable and immediately viable next step.
>
> > **Section 2, for instance, seems pretty important considering their claimed contribution but looks not fully fleshed out.**
>
> As discussed above, there are many things to cover in our 9 pages. We agree that Section 2 is particularly important, and we elected to briefly introduce these concepts there before unpacking them in a more concrete context in Section 3. We have updated Section 2 to make this flow clearer to the reader from the start.
>
> > **Maybe Figure 1 can be improved to better relay the idea of time-step inputs?**
>
> We interpret this comment to mean that it should be made more clear that the message `z_t` depicted in Figure 1 should be more clearly depicted as an input (like `x_t`). We agree, and have updated the figure in the main text. We also introduced a new expanded Figure S1 in the supplement that relays these ideas in more detail, which we hope clarifies the core design.
>
> > **Q1) …this seems to be a big claim …**
>
> We agree — this strong claim slipped in from an earlier draft of the paper. We have amended this.
>
> > **Q2) Section 2: I would elaborate this section more. Especially “faithful” since that seems to be multiple desiderata in and of itself and vague. Also maybe add in-line citations.**
>
> We find that faithfulness is best understood in the context of decoders, specifically in how they fail to be faithful. This is discussed in context at the end of Section 3 (with 10 in-line citations). Nonetheless, we agree that it would be useful for the reader to at least be aware that this fuller explanation is forthcoming and have updated Section 2 accordingly.
>
> > **Typos and Writing suggestions**
>
> We agree with all the reviewer’s suggestions and have updated the paper accordingly. In particular, we agree that the introduction could mention “feed[ing] the model’s outputs as inputs” as a key step of CST.
>
> ---
>
> Please let us know if these changes to the paper are not what was intended, as we are eager to make our paper as clear as possible. Given that the primary objection to the work was due to the presentation, we would ask that the reviewer reconsider their recommendation that this paper be rejected in light of these revisions, such that the contributions of this work can reach the larger NeurIPS audience.

---

> > ### Comment · Reviewer_AH1b · 2022-08-09
> > **thank you for your updates**
> >
> > Thank you authors for engaging with my comments and taking steps to improve the clarity of the paper overall. I will increase the presentation score accordingly.

---

### Author Response · Authors · 2022-08-02
**Updated manuscript**

We greatly appreciate the reviewers’ valuable and insightful comments and suggestions.

We have edited the paper accordingly, and believe it has improved significantly. We hope each reviewer considers adjusting their score upward in light of these improvements.

Note that, given the edits, our page count goes slightly over 9 pages. We intend to address this as the reviewers reach consensus.

---

> ### Author Response · Authors · 2022-08-09
> **Updated manuscript (Rev 2)**
>
> As promised, we have amended our previous submission to return the page count to 9 pages. This has involved merely minor changes to the text.

---

### Meta-Review · Area_Chair_Dsa4 · 2022-08-21

**Recommendation:** Accept
**Confidence:** Less certain

**Metareview:**

This paper proposes causal self-talk (CST) as a means to obtain more explainable AI systems. The work lists a set of desiderata for explainable AI and argues that CST satisfies this set. The paper is well written and the experimental results, although in a toy setting called "DaxDucks" are reasonably convincing. The reviewers lean towards acceptance, with Reviewer g2gH in particular strongly advocating for the work.

I like that this approach is generally applicable in principle, and I strongly dislike that it is only showcased on one (non-open source, toy) task. The other qualm I have is that the introduction reads way too much as if the authors came up with this all on their own, and I strongly encourage them to pay proper respect to prior work in this field, rather than stashing all that work away as a long enumeration in the related work section, which most people will gloss over.

**Award:**

No

---

### Decision · Program_Chairs · 2022-09-14

Accept